# Diagnostic ability of confocal scanning ophthalmoscope for the detection of concurrent retinal disease in eyes with asteroid hyalosis

**Su Hwan Park**[1], **Ji Hyoung Chey**[2], **Jun Heo**[3], **Kwang Eon Han**[1], **Sung Who Park**[3], **Iksoo Byon**[3], **Han Jo Kwon**[3]*

1 Department of Ophthalmology, Research Institute for Convergence of Biomedical Science and Technology, Pusan National University Yangsan Hospital, Yangsan-si, Gyeongsangnam-do, South Korea, 2 Department of Ophthalmology, Ulsan University Hospital, University of Ulsan College of Medicine, Ulsan, South Korea, 3 Department of Ophthalmology, Biomedical Research Institute, Pusan National University Hospital, Pusan National University School of Medicine, Busan, South Korea

* snowcjh@hanmail.net

**Data Availability Statement:** All relevant data are within the manuscript and its Supporting Information files.

## Abstract

### Purpose

To compare the diagnostic capacity of a color fundus camera (CFC), ultra-wide-field bicolor confocal scanning laser ophthalmoscope (BC-cSLO; OPTOS), and true-color confocal scanning ophthalmoscope (TC-cSO; EIDON) in detecting coexisting retinal diseases in eyes with asteroid hyalosis (AH).

### Methods

The medical records of consecutive patients with AH who were referred to a tertiary hospital for subsequent assessment by a vitreoretinal specialist were retrospectively reviewed. Fundus images obtained simultaneously using CFC, BC-cSLO, and TC-cSO were classified into four grades based on their obscuration by asteroid bodies. The proportion of Grade 1 images (minimal obscuration group) was assessed for each imaging modality. The diagnostic and screening abilities for concurrent retinal diseases were compared in terms of the accuracy and sensitivity of each device.

### Results

Among the 100 eyes with AH, 76 had coexisting retinal diseases, such as diabetic retinopathy (DR), retinal vascular occlusion, age-related macular degeneration, epiretinal membrane, and retinitis pigmentosa. TC-cSO had the highest ratio of Grade 1 images (94%, P<0.001), followed by CFC (67%) and BC-cSLO (63%). CFC and BC-cSLO exhibited a 5.3-fold higher rate of significant obscuration than TC-cSO (P<0.001, 95% confidence intervals = 2.4~11.6 folds). TC-cSO demonstrated the highest accuracy and sensitivity (95% and 81%, respectively) compared with CFC (89% and 43%, respectively) and BC-cSLO (89%

**Funding:** The authors have not received a specific grant for this research from any funding agency in the public, commercial, or not-for-profit sectors.

**Competing interests:** No conflicting relationship exists for any author.

and 39%, respectively) for all retinal diseases. BC-cSLO showed the best performance for DR diagnosis.

## Conclusions

TC-cSO images showed minimal obscuration and a superior ability for diagnosing retinal diseases accompanied by AH over other imaging devices. TC-cSO can be a valuable alternative screening tool for detecting retinal diseases when AH impedes fundus imaging.

## Introduction

Asteroid hyalosis (AH) is a degenerative disease characterized by the presence of star-like asteroid bodies formed by calcium and fatty acid compounds floating around the vitreous humor [1, 2]. AH is more prevalent in males and typically occurs unilaterally [3]. The prevalence of AH is reported to be approximately 1.2% among individuals aged ≥55 years. The prevalence increases with age, from 0.3% in individuals aged <40 years to 3.1% in individuals aged ≥80 years [3, 4]. Asteroid bodies, typically located in the anterior vitreous humor, do not impair visual acuity. AH is a benign disease that rarely requires vitrectomy [3, 5].

AH can be accompanied by vision-threatening retinal diseases. The diagnosis of retinal diseases using fundus examination or a color fundus camera (CFC) is often hindered if asteroid bodies obscure the macula [5]. Wright et al. [6] reported that some fundus photographs could not be interpreted during screening for diabetic retinopathy (DR) because of dense asteroid bodies. Alternative methods, such as fluorescein angiography, optical coherence tomography (OCT), OCT angiography, B-scan ultrasonography, and multicolor fundus imaging have been proposed [7–9]. However, these methods often face challenges such as time-consuming processes and technical difficulties in investigating the color and spatial distribution of retinal lesions. Furthermore, there are significant limitations to the adoption of screening equipment owing to its high cost and requirement for skilled operators.

Retinal imaging devices based on confocal scanning ophthalmoscopy (cSOs), such as ultra-wide-field (UWF) bicolor confocal scanning laser ophthalmoscopes (BC-cSLOs) and true-color confocal scanning ophthalmoscopes (TC-cSOs), have been used in clinical settings [10, 11]. cSOs, a type of confocal microscope, utilizes a laser or a light-emitting diode to illuminate specific retinal regions. Light reflected from the retina passes through a narrow aperture or slit and only the image in the desired focal plane is detected. Scanning optics were used to reconstruct two-dimensional retinal images. This technique enhances the contrast of retinal images by blocking light from planes other than the focal plane [10, 11].

To the best of our knowledge, there has been no research on the clinical efficacy of cSOs in diagnosing retinal diseases co-occurring with AH. In this study, we aimed to evaluate the accuracy of each device in detecting the accompanying retinal diseases and compare the overall diagnostic capability of retinal imaging devices by assessing the degree of asteroid body obscuration.

## Materials and methods

This retrospective and comparative study was approved by the Institutional Review Board (IRB) of Pusan National University Yangsan Hospital (PNUYH) (IRB No. 05-2023-111) and conducted in accordance with the Declaration of Helsinki. The IRB of PNUYH approved a

request to waive the documentation of informed consent for this retrospective research. We accessed the Electronic Medical Records (EMR) from June 26, 2023, to September 29, 2023. The EMR and Digital Imaging and Communications in Medicine image sets of the patients were anonymized, and we could not identify individual participants during or after data collection. Additionally, this manuscript was written in strict adherence to the STARD 2015 guidelines [12].

This study included patients referred to the PNUYH between January 2021 and December 2022 for initial evaluation by a vitreoretinal specialist. These patients were either suspected of having retinal disease accompanied by AH or were unable to undergo fundus examination because of densely packed asteroid bodies. These patients were confirmed to have AH on their first visit through a dilated fundus examination. At the next visit, their affected eyes were imaged using TRC-NW200 CFC (Topcon Corporation, Tokyo, Japan), followed by additional imaging with Optos California® UWF™ BC-cSLO (Optos PLC, Dunfermline, UK) and Eidon AF™ TC-cSO (CenterVue, Padova, Italy) in a non-mydriatic state. Each examination was performed by a skilled examiner to ensure a minimum gap of one minute between imaging sessions. Patients with conditions other than AH that could affect the quality of fundus photography, such as corneal opacity, grade ≥4 nuclear opacity, or grade ≥2 posterior subcapsular opacity according to the Lens Opacities Classification System III [13], or vitreous hemorrhage were not included in the study. Additionally, patients with retinal or choroidal detachment that interfered with focusing on the entire macula, those whose pupils did not dilate to at least 4 mm under low light (5 lx), and those with involuntary eye movements causing overlapping images were excluded from the study.

Patient demographics and ocular characteristics were collected by reviewing medical records, including age and sex, along with underlying diseases, such as diabetes, hypertension, dyslipidemia, best-corrected visual acuity (BCVA), intraocular pressure, and the presence and type of retinal disease. After anonymization, the corresponding author (H.J.K.) saved each fundus photograph acquired using CFC, UWF BC-cSLO, and TC-cSO, in the order of patient visits. A unique random number was generated using Python software (version 3.9.16; Python Software Foundation, Wilmington, Delaware, USA) and assigned to each image filename to reduce the high correlation between adjacent photographs.

Grade assessment and diagnosis of the fundus photographs were performed in the same environment using a 24-inch 8-bit monitor with a resolution of 3840×2160. All retinal images were delivered to ophthalmologists in their original form, without cropping or resizing, and in TIFF file format. The ophthalmologists had no time constraints for interpretation and could adjust the image brightness and contrast using image editing software (Photoshop CS3 ver. 10.0; Adobe Systems, Mountain View, CA, USA). To maintain independent assessments, each ophthalmologist worked individually and submitted their findings to the corresponding author.

For grade assessment, we categorized the degree of obscuration of the posterior pole by the asteroid bodies into four grades according to the classification method of Rani et al. [9]: Grade 1, optic disc and second-order vessels visible; Grade 2, optic disc and first-order vessels visible; Grade 3, hazy appearance of the optic disc, first-order vessels not clearly visible; and Grade 4, posterior pole, including the optic disc, completely obscured (**Fig 1**).

Two ophthalmologists (S.H.P. and J.H.C.) conducted the grade assessments, with the corresponding author making the final decision in cases of disagreement. Images were divided into two groups: a minimal obscuration group (Grade 1) and a significant obscuration group (Grades 2–4). We analyzed the ratios of these groups among the devices to identify any statistically significant differences.

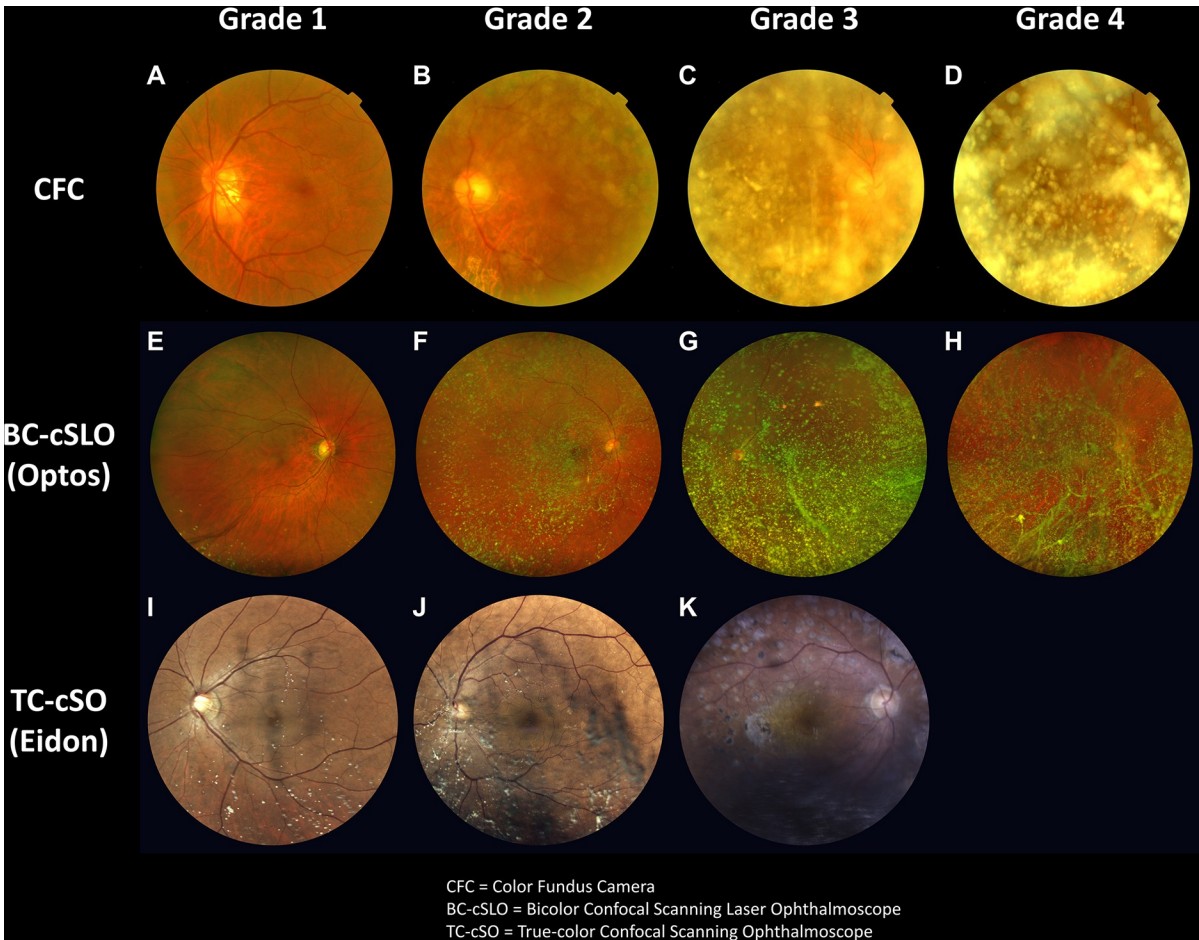

**Fig 1. Grade assessment based on the degree of posterior retina obscuration by asteroid bodies.** (**A–D**) Top row: Fundus photographs of patients with asteroid hyalosis (AH) captured using a color fundus camera (CFC). (**E–H**) Second row: Images of patients with AH taken with an ultrawide-field bicolor confocal scanning laser ophthalmoscope (BC-cSLO). (**I–K**) Third row: AH patient photos captured using a true-color confocal scanning ophthalmoscope (TC-cSO). Grading was assigned as follows: Grade 1, clearly visible optic disc and second-order vessels; Grade 2, for a visible optic disc and first-order vessels; 3, hazy optic disc with first-order vessels not clearly visible; and Grade 4, posterior pole and optic completely obscured due to dense asteroid bodies. Each column corresponds to the same grade, which increases from left to right. Notably, no images taken with TC-cSO were classified as Grade 4.

Using the confusion matrix for performance evaluation offers detailed classification outcomes, while deriving key metrics like accuracy, precision, sensitivity, and F1 score provides a comprehensive evaluation of performance across different classes. Diagnostic and screening abilities for concurrent retinal diseases in patients with AH were investigated using the following four-step process: First, based on the medical records and three fundus images, the corresponding author diagnosed concurrent retinal disease for each retinal image as a reference diagnosis. The corresponding author created a list of potential retinal diseases including DR, dry and wet age-related macular degeneration (AMD), retinal vein occlusion (RVO), epiretinal membrane (ERM), and retinitis pigmentosa. Third, four independent ophthalmologists (S.H. P., J.H.C., J.H., and K.E.H.) recorded possible retinal diseases accompanying AH as diagnostic tests, based solely on fundus photographs taken with CFCs, UWF BC-cSLOs, and TC-cSOs. The diagnostic test was continued until each ophthalmologist interpreted all fundus photographs. Finally, the corresponding author compared the diagnoses of the four ophthalmologists against the reference diagnosis and constructed confusion matrices for each

ophthalmologist's assessment. These matrices were aggregated to calculate the accuracy and sensitivity of each imaging device for each disease and to evaluate their diagnostic capabilities. In addition, we compared the diagnostic abilities of the three imaging devices based on the obscuration grade of the asteroid bodies observed in the CFC. Therefore, we investigated the equipment that would be effective in screening for background retinal disease in cases of severe-grade AH in CFC.

To estimate the sample size, we investigated the fundus photographs of AH patients from September to December 2020 using the same inclusion and exclusion criteria based on their medical records. We examined the rate of Grade 1 AH using each imaging device. The findings were as follows: 19 out of 34 eyes in the CFC images, 23 out of 34 eyes in the UWF BC-cSLO images, and 28 out of 34 eyes in the TC-cSO images. According to an analysis using G*Power version 3.1.9.7 [14], we could calculate that a minimum of 82 subjects were required in TC-cSO to confirm the statistical difference in the ratio of minimal obscuration. To compare the degree of obscuration by asteroid bodies among the devices, we employed Chi-square and Fisher's exact tests. Fleiss' kappa values for each disease were calculated to assess the inter-rater reliability (IRR) among ophthalmologists, with values interpreted as poor ($<0.40$), moderate ($0.40-0.75$), and excellent ($>0.75$) agreement [15]. The kappa values were computed using the statmodels version 0.13.2 library in Python. All other statistical analyses were performed using SPSS for Windows (version 23.0; SPSS Inc., Chicago, IL, USA). Statistical significance was set at P-value $< 0.05$.

## Results

Of the 101 patients (119 eyes) diagnosed with AH, 89 patients (100 eyes) met the inclusion and exclusion criteria detailed in **S1 Fig**. Patients underwent fundus imaging with the CFC and two types of cSO. Among them, 52.8% (47 patients) were male, with an average age of 70.6 ±11.1 years. Forty-three (48.3%) patients had type 2 diabetes, 50 (56.2%) had hypertension, and 13 (14.6%) had dyslipidemia. The mean BCVA at the second visit was 0.303±0.405 log-MAR, and the mean intraocular pressure was 15.1±2.9 mmHg. Concurrent retinal disease with AH was observed in 76.0% of the eyes (76 eyes) (**Table 1**). Twenty-seven patients with AH had DR or Wet AMD.

Grade assessment of fundus photographs by imaging devices showed the following: CFC had 67.0% of eyes in Grade 1, 26.0% in Grade 2, 6.0% in Grade 3, and 1.0% in Grade 4; UWF BC-cSLO had 63.0% in Grade 1, 21.0% in Grade 2, 14.0% in Grade 3, and 2.0% in Grade 4; and TC-cSO had 94.0% in Grade 1 and 6.0% distributed across Grades 2 and 3, with none in Grade 4. There was a significant difference between the devices (P<0.001), with TC-cSO showing statistically fewer cases of significant obscuration (6.0%) than CFC and UWF BC-cSLO (35.0%). The significant obscuration was 5.3 times greater with CFC and UWF BC-cSLO than with TC-cSO (risk ratio = 5.3, 95% confidence intervals = 2.4~11.6, P<0.001) (**Table 2**).

The average clinical experience of the four ophthalmologists was 4.3±2.3 years. The Fleiss' Kappa values for the IRR ranged from 0.537 to 0.809, indicating moderate-to-excellent agreement among ophthalmologists across all diseases. In the analysis of IRR for each specific device-disease combination, the lowest Fleiss' Kappa values for DR and wet AMD were observed with CFC. For the other diseases, the lowest values were noted for UWF BC-cSLO. Notably, TC-cSO did not exhibit the lowest Fleiss' Kappa value for any disease (**Table 3**).

The four ophthalmologists diagnosed six concurrent retinal diseases using 100 fundus photographs for three imaging devices (CFC, BC-cSLO, and TC-cSLO), and the diagnostic performance of each imaging device was evaluated using confusion matrices. **Table 4** summarizes the diagnostic capabilities of each imaging device. TC-cSO demonstrated the best diagnostic

**Table 1. Baseline characteristics for eyes with asteroid hyalosis.**

| Factors | Total Eyes ($N_e$ = 100) and Patients ($N_p$ = 89) |
|---|:---:|
| **Baseline Parameters** | |
| Age (years) | 70.6±11.1 |
| Right/Left ($N_e$, %) | 54 (54.0%)/46 (46.0%) |
| Male/Female ($N_p$, %) | 47 (52.8%)/42 (47.2%) |
| Bilateral/Unilateral ($N_p$, %) | 11 (12.4%)/78 (87.6%) |
| Phakia/Pseudophakia ($N_e$, %) | 63 (63.0%)/37 (37.0%) |
| BCVA (logMAR) | 0.303±0.405 |
| IOP (mmHg) | 15.1±2.9 |
| **Underlying Disease ($N_p$, %)** | 68 (76.4%) |
| Diabetes mellitus ($N_p$, %) | 43 (48.3%) |
| Hypertension ($N_p$, %) | 50 (56.2%) |
| Dyslipidemia ($N_p$, %) | 13 (14.6%) |
| **Concurrent Retinal Disease ($N_e$, %)** | 76 (76.0%) |
| DR ($N_e$, %) | 20 (20.0%) |
| Dry AMD ($N_e$, %) | 31 (31.0%) |
| Wet AMD ($N_e$, %) | 9 (9.0%) |
| RVO ($N_e$, %) | 6 (6.0%) |
| ERM ($N_e$, %) | 33 (33.0%) |
| RP ($N_e$, %) | 3 (3.0%) |
| Two retinal diseases[a] ($N_e$, %) | 18 (18.0%) |
| Three retinal diseases[b] ($N_e$, %) | 4 (4.0%) |

AMD, Age-related macular degeneration; BCVA = Best-corrected visual acuity; DR = Diabetic retinopathy; ERM = Epiretinal retinal membrane; IOP = Intraocular pressure; logMAR, Logarithm of the minimum angle of resolution; $N_e$ = Number of eyes; $N_p$ = Number of patients; RP = Retinitis pigmentosa; RVO = Retinal vein occlusion.
[a] Two retinal diseases and [b] Three retinal diseases represent the number of eyes diagnosed with multiple diseases among the listed concurrent retinal disease (DR, Dry AMD, Wet AMD, RVO, ERM, and RP).

ability for all diseases, with an accuracy of 94.6% and sensitivity of 80.9%. Additionally, its F1 score exceeded 0.7 for all diseases. Notably, the sensitivity of the TC-cSO outperformed other devices in almost all categories: RVO (87.5%), retinitis pigmentosa (83.3%), dry AMD (71.0%), wet AMD (63.9%), and ERM (84.6%). The diagnostic ability of UWF BC-cSLO was particularly effective for DR accompanied by AH, achieving an accuracy of 98.0% and a sensitivity of 93.8%.

We categorized the obscuration assessed using CFC into two groups (minimal and significant obscuration) and compared the diagnostic abilities of UWF BC-cSLO and TC-cSO for these two groups (**S1 Table**). In the minimal obscuration group, the diagnostic accuracy ranged from 89.3% to 95.1% across the three devices, indicating favorable accuracy. However, the sensitivity was 51.4% for CFC, similar to 45.0% for BC-cSLO, but notably lower than the 83.6% achieved by TC-cSO. In the significant obscuration group, the sensitivities decreased to 24.2% and 25.8% for both CFC and BC-cSLO, respectively, whereas TC-cSO maintained a higher sensitivity of 75.0%. **Fig 2** illustrates these differences in diagnostic capabilities under varying grades of obscuration in CFC images.

## Discussion

In this study, approximately 75% of the patients referred to a tertiary hospital for AH had concurrent retinal diseases. ERM and dry AMD are the two most common retinal diseases

**Table 2. Comparison of the grade assessment in fundus photographs by three imaging devices based on the degree of obscuration by asteroid bodies.**

| Imaging Device | Grade | | | |
|---|---|---|---|---|
| | Grade 1 | Grade 2 | Grade 3 | Grade 4 |
| CFC (N) | 67 | 26 | 6 | 1 |
| BC-cSLO (N) | 63 | 21 | 14 | 2 |
| TC-cSO (N) | 94 | 3 | 3 | 0 |
| P-Value | <0.001 | | | |
| | Minimal Obscuration | | Significant Obscuration | |
| CFC (N) (N) | 67 | | 33 | |
| BC-cSLO (N) | 63 | | 37 | |
| TC-cSO (N) | 94 | | 6 | |
| P-Value | <0.001 | | | |
| | Minimal Obscuration | | Significant Obscuration | |
| CFC and BC-cSLO (N, %) | 130 (65.0) | | 70 (35.0) | |
| TC-cSO (N, %) | 94 (94.0) | | 6 (6.0) | |
| P-Value | <0.001 (Risk ratio = 5.3 and 95% confidence intervals = 2.4~11.6) | | | |

BC-cSLO, Bicolor confocal scanning laser ophthalmoscope; CFC, Color fundus camera; TC-cSO, True-color confocal scanning ophthalmoscope.

Grade of obscuration was defined as minimal for Grade 1 and significant for Grades 2–4.

associated with AH, accounting for over 60% of cases. DR and wet AMD, the major retinal diseases responsible for adult vision loss, occur in approximately 30% of patients with AH. Compared to other devices, TC-cSO imaging was less affected by asteroid body obscuration, leading to the best diagnostic ability and highest agreement among ophthalmologists. Except for concurrent DR with AH, the TC-cSO demonstrated superior diagnostic capabilities for all other retinal diseases compared with alternative devices. This improvement became more evident in patients as the obscuration by asteroid bodies increased in color fundus photography.

Visual acuity is seldom affected in patients with AH, even in the presence of densely packed asteroid bodies that interfere with fundus examinations and imaging [5]. This study also found no difference in baseline BCVA between the minimal and significant obscuration groups. Particles in the vitreous humor can scatter light, reduce visual quality, and impair contrast sensitivity [16, 17]. However, particles with smooth surfaces exhibit minimal light scattering [17]. The surface of asteroid bodies comprises fibrillar collagen-like structures in a lamellar arrangement at regular nanometer intervals, leading to an optically smooth surface [18]. Therefore,

**Table 3. Inter-rater reliability in the interpretation of retinal diseases accompanied by asteroid hyalosis across three imaging devices.**

| Imaging Device | Concurrent Retinal Disease | | | | | |
|---|---|---|---|---|---|---|
| | DR | Dry AMD | Wet AMD | RVO | ERM | RP |
| CFC | 0.675* | 0.610 | 0.427* | 0.862 | 0.545 | 0.662 |
| BC-cSLO | 0.883 | 0.468* | 0.551 | 0.693* | 0.284* | 0.659* |
| TC-cSO | 0.840 | 0.498 | 0.566 | 0.701 | 0.714 | 0.782 |
| All Devices | 0.809 | 0.551 | 0.537 | 0.747 | 0.674 | 0.712 |

AMD, Age-related macular degeneration; BC-cSLO, Bicolor confocal scanning laser ophthalmoscope; CFC, Color fundus camera; DR, Diabetic retinopathy; ERM, Epiretinal membrane; IRR, Inter-rater reliability; RP, Retinitis pigmentosa; RVO, Retinal vein occlusion; TC-cSO, True-color confocal scanning ophthalmoscope.

All numbers represent Fleiss's Kappa values, with values interpreted as poor (<0.40), moderate (0.40–0.75), and excellent (>0.75) agreement, respectively.

* values represent the lowest Fleiss's Kappa values among the imaging devices for each retinal disease.

**Table 4. Comparative analysis of the diagnostic ability of three imaging devices for concurrent retinal diseases with asteroid hyalosis.**

| Imaging Device | Concurrent Retinal Disease | | | | | | |
|---|---|---|---|---|---|---|---|
| **CFC** | **DR** | **Dry AMD** | **Wet AMD** | **RVO** | **ERM** | **RP** | **All Diseases** |
| TP (N) | 52 | 61 | 10 | 19 | 28 | 5 | 175 |
| TN (N) | 313 | 261 | 362 | 372 | 265 | 387 | 1960 |
| FP (N) | 7 | 15 | 2 | 4 | 3 | 1 | 32 |
| FN (N) | 28 | 63 | 26 | 5 | 104 | 7 | 233 |
| Accuracy (%) | 91.3 | 80.5 | 93.0 | 97.8 | 73.3 | 98.0 | 89.0 |
| Sensitivity (%) | 65.0 | 49.2 | 27.8 | 79.2 | 21.2 | 41.7 | 42.9 |
| Specificity (%) | 97.8 | 94.6 | 99.5 | 98.9 | 98.9 | 99.7 | 98.4 |
| Precision (%) | 88.1 | 80.3 | 83.3 | 82.6 | 90.3 | 83.3 | 84.5 |
| F1 score | 0.748 | 0.610 | 0.417 | 0.809 | 0.344 | 0.556 | 0.569 |
| **BC-cSLO** | **DR** | **Dry AMD** | **Wet AMD** | **RVO** | **ERM** | **RP** | **All Diseases** |
| TP (N) | 75 | 29 | 18 | 18 | 12 | 7 | 159 |
| TN (N) | 317 | 270 | 360 | 366 | 265 | 386 | 1964 |
| FP (N) | 3 | 6 | 4 | 10 | 3 | 2 | 28 |
| FN (N) | 5 | 95 | 18 | 6 | 120 | 5 | 249 |
| Accuracy (%) | 98.0 | 74.8 | 94.5 | 96.0 | 69.3 | 98.3 | 88.5 |
| Sensitivity (%) | 93.8 | 23.4 | 50.0 | 75.0 | 9.1 | 58.3 | 39.0 |
| Specificity (%) | 99.1 | 97.8 | 98.9 | 97.3 | 98.9 | 99.5 | 98.6 |
| Precision (%) | 96.2 | 82.9 | 81.8 | 64.3 | 80.0 | 77.8 | 85.0 |
| F1 score | 0.949 | 0.365 | 0.621 | 0.692 | 0.163 | 0.667 | 0.534 |
| **TC-cSO** | **DR** | **Dry AMD** | **Wet AMD** | **RVO** | **ERM** | **RP** | **All Diseases** |
| TP (N) | 74 | 88 | 23 | 21 | 114 | 10 | 330 |
| TN (N) | 313 | 254 | 358 | 372 | 256 | 387 | 1940 |
| FP (N) | 7 | 22 | 6 | 4 | 12 | 1 | 52 |
| FN (N) | 6 | 36 | 13 | 3 | 18 | 2 | 78 |
| Accuracy (%) | 96.8 | 85.5 | 95.3 | 98.3 | 92.5 | 99.3 | 94.6 |
| Sensitivity (%) | 92.5 | 71.0 | 63.9 | 87.5 | 86.4 | 83.3 | 80.9 |
| Specificity (%) | 97.8 | 92.0 | 98.4 | 98.9 | 95.5 | 99.7 | 97.4 |
| Precision (%) | 91.4 | 80.0 | 79.3 | 84.0 | 90.5 | 90.9 | 86.4 |
| F1 score | 0.919 | 0.752 | 0.708 | 0.857 | 0.884 | 0.870 | 0.835 |

AMD, Age-related macular degeneration; BC-cSLO, Bicolor confocal scanning laser ophthalmoscope; CFC, Color fundus camera; DR, Diabetic retinopathy; ERM, Epiretinal membrane; FN, False Negative; FP, False Positive; RP, Retinitis pigmentosa; RVO, Retinal vein occlusion; TC-cSO, True-color confocal scanning ophthalmoscope; TN, True Negative; TP, True Positive.

TP, TN, FP, and FN represent the values of the confusion matrix for each imaging device and disease. TP refers to the number of positive cases that were correctly identified. TN refers to the number of negative cases that were correctly identified. FP refers to the number of negative cases that were incorrectly identified as positive. FN refers to the number of positive cases that were incorrectly identified as negative.

Accuracy (%) = (TP + TN) / (TP + TN + FP + FN) × 100

Sensitivity (%) = TP / (TP + FN) × 100

Specificity (%) = TN / (TN + FP) × 100

Precision (%) = TP / (TP + FP) × 100

F1 score = 2 × (Precision × Sensitivity) / (Precision + Sensitivity)

unlike collagen vitreous floaters with irregular surfaces, asteroid bodies do not cause ocular discomfort [5].

It is also associated with systemic and retinal diseases. Diabetes mellitus, hyperlipidemia, arterial hypertension, atherosclerosis, and hypercalcemia are commonly observed in patients with AH [5, 19–23]. There is no definitive evidence that specific retinal diseases are more

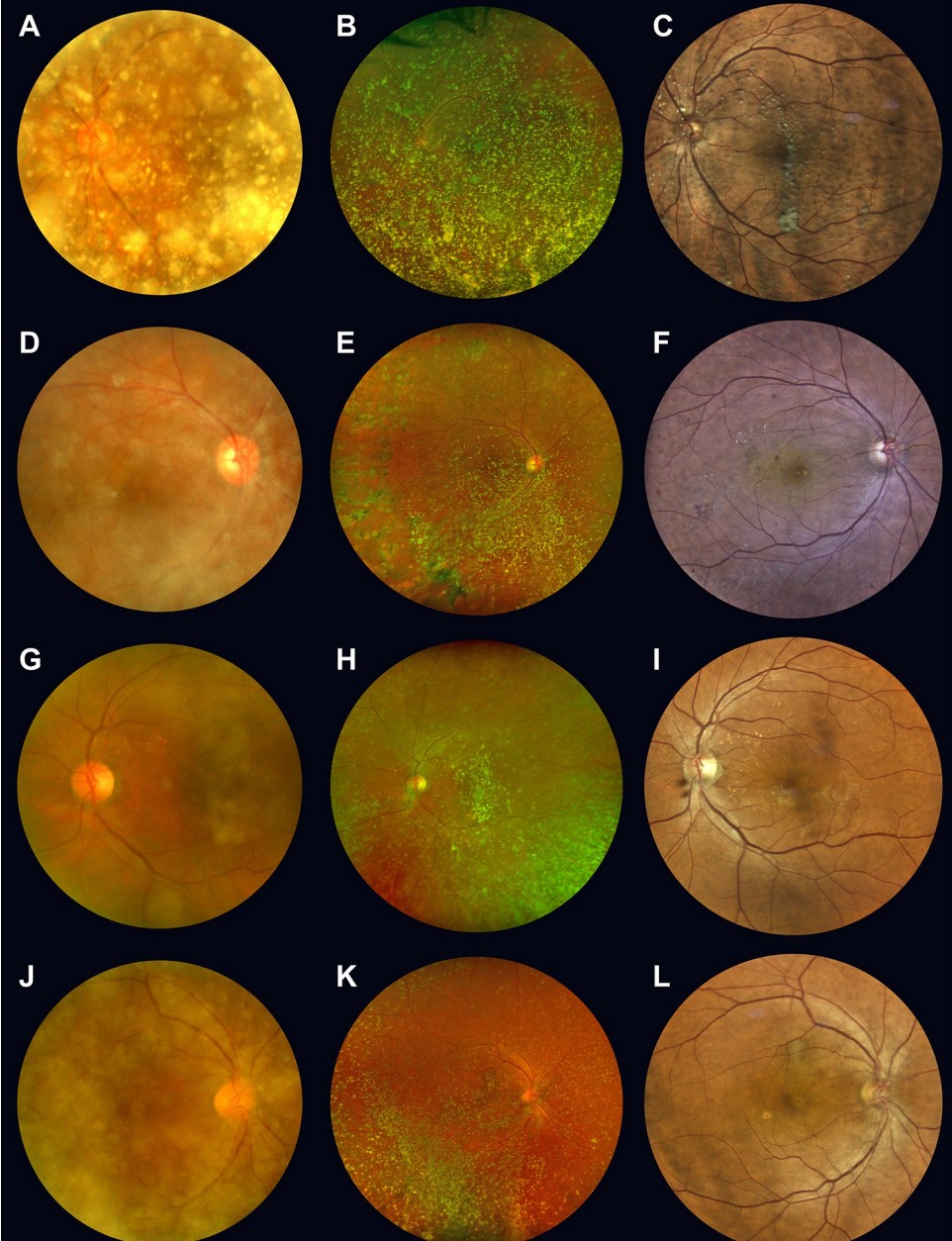

**Fig 2. Differences in fundus images obtained by three imaging devices in patients with asteroid hyalosis.** Each row consists of fundus photographs taken from the same patient with asteroid hyalosis (AH). (**A**) Grade 3 obscuration in color fundus camera (CFC) shows dense asteroid bodies obscuring the optic disc and macula. (**B**) Bicolor confocal scanning laser ophthalmoscope (BC-cSLO) also displays similar obscuration, providing no additional information on the macular status. (**C**) True-color confocal scanning ophthalmoscope (TC-cSO) reveals reduced asteroid bodies, except those shadows on the macula. No retinal disorders are evident. (**D**) Similar to **A**, Grade 3 obscuration in CFC is evident, and a single retinal hemorrhage is barely detectable. (**E**) BC-cSLO shows peripheral retinal photocoagulation scars, indicative of diabetic retinopathy, but asteroid bodies still impede the macular region. (**F**) TC-cSO uncovers microaneurysms and hard exudates on the macula, leading to a unanimous diagnosis of diabetic retinopathy. (**G**) CFC displays multiple drusens with asteroid bodies, diagnosed as dry AMD with AH. (**H**) BC-cSLO reveals faint retinal folding near the fovea, diagnosed by one examiner as an epiretinal membrane. (**I**) TC-cSO shows membrane reflex and retinal folding, which are previously obscured in images **G** and **H**, clearly suggesting ERM and dry AMD with AH. (**J**) Grade 3 obscuration by asteroid bodies in CFC. (**K**) BC-cSLO exhibits a faint, round pigment epithelial detachment in the upper parafovea. (**L**) TC-cSO reveals well-circumscribed hemorrhagic pigment epithelial detachment due to wet AMD.

common in patients with AH; however, several cases of DR can coexist in patients with bilateral AH [20, 24]. Retinitis pigmentosa and Alström syndrome are also associated with AH [25, 26]. Moreover, three large-sample retrospective studies demonstrated an increased prevalence of AH with old age [3, 27, 28]. Old age increases the incidence of degenerative retinal diseases that affect the macula, including DR, RVO, and AMD [29–31]. Therefore, ophthalmologists should be aware of the risk of hidden underlying retinal diseases that can threaten visual function in elderly patients with AH that obscures the macula, even if their recent visual acuity is favorable.

A national screening program conducted in Walsall, West Midlands, UK, involving 260,000 individuals found that DR could not be confirmed using CFC in 0.4% of patients with diabetes owing to interference from asteroid bodies [6]. Furthermore, interpreting color fundus photographs was impossible in 8.5% of cases owing to the high density of asteroid bodies, emphasizing the necessity of retesting these patients with alternative methods [6]. Given the high number of diabetes cases in South Korea, estimated to reach 6 million in 2022, approximately 14, 000 patients would fail DR screening owing to ungradable fundus photography [6, 32]. Considering other degenerative retinal diseases in patients with AH, the need for alternative diagnostic methods may be even greater.

This study showed a significant difference in the sensitivities of the three imaging devices for detecting retinal diseases in patients with AH, as detailed in **Table 4**. TC-cSO demonstrated a notably high sensitivity, exceeding 80%, for all retinal diseases. In contrast, CFC and UWF BC-cSLO exhibited considerably lower sensitivities (approximately 40%). This finding suggests that the CFC and UWF BC-cSLO may not be adequate primary screening tools for retinal diseases in patients with AH. In particular, when severe obscuration was evident in the CFC images, the sensitivity dropped to approximately 25% for both CFC and UWF BC-cSLO. TC-cSO maintained a relatively high sensitivity of 75%. Given the predominant use of CFCs in health examination centers by primary healthcare providers, this study recommends that TC-cSO be considered as an alternative for screening retinal diseases in patients with AH, especially when dense asteroid bodies hinder the interpretation of conventional color fundus photography.

UWF BC-cSLO can image the retina over a 200˚ range using green and red lasers to produce pseudocolor images [10]. Compared to CFC, asteroid bodies imaged with UWF BC-cSLO exhibited less reflection [33, 34]. Nevertheless, our study found that the F1 scores of the UWF BC-cSLO were usually < 0.7. Interestingly, it scored higher for DR detection, exceeding 0.9. UWF BC-cSLO imaging has proven effective in identifying peripheral lesions in diseases such as DR, RVO, and retinitis pigmentosa. However, its diagnostic ability is lower for macular diseases such as AMD and ERM. The UWF BC-cSLO has a lower spatial resolution of 20 pixels/˚, in contrast to TC-cSO's higher resolution of 60 pixels/˚. Consequently, UWF BC-cSLO offers less detailed views of macular lesions than TC-cSO, potentially affecting its effectiveness in the diagnosis of macular diseases. Studies comparing the chromaticity of TC-cSO with that of CFC and other UWF fundus cameras have revealed that TC-cSO images display a wider and more balanced color range [35], whereas CFC images often appear reddish [11]. Therefore, the difference in diagnostic capability in this study might stem from differences in the spatial resolution and color balance of the three imaging devices.

The confocal effect of TC-cSO effectively minimized interference from asteroid bodies in fundus photographs, leading to less obscuration compared to images from CFC and UWF BC-cSLO. The TC-cSO confocal effect restricts information to deeper layers, such as the choroidal vessels in the red channel. In contrast, CFC and UWF BC-cSLO incorporate this deeper information into the red channel. Consequently, microvascular abnormalities, which may appear faint in CFC images, were more distinctly visible in the TC-cSO images [35]. These advantages

can enhance TC-cSO's interreader reliability (IRR), sensitivity, and accuracy. Interestingly, despite having a smaller field of view (FOV), TC-cSO surpassed the UWF BC-cSLO in terms of accuracy and sensitivity. This suggests that enhanced spatial resolution and confocal effects might be more important than a larger FOV for the accurate diagnosis of retinal diseases in patients with AH. Given the cost of each device, TC-cSO may be a more cost-effective option for diagnosing retinal diseases concurrent with AH.

This study has several limitations, including a small sample size and a limited number of retinal diseases. If the assessment covers retinal diseases such as rhegmatogenous retinal detachment and acute retinal necrosis originating from the peripheral retina, BC-cSLO's diagnostic capabilities of UWF BC-cSLO may be rated higher. Furthermore, we could not compare images obtained using other fundus cameras and cSO imaging devices. We attempted to include patients with various retinal diseases, which allowed us to compare the overall diagnostic abilities of the three imaging devices. Our study population, which was primarily referred to a tertiary hospital for suspected retinal abnormalities, may have exhibited a selection bias. This bias could lead to results that are not fully representative of the general population visiting primary healthcare centers. This study did not distinguish between the types of referable retinal diseases associated with AH; instead, it focused on the presence of retinal disease. Nonetheless, this study could be suitable for the efficient use of time and resources to explore the accuracy and sensitivity of patients at risk of retinal disease before conducting research on referable retinal diseases in a broader population. The grade assessment of AH obscuration was classified not quantitatively but qualitatively. In the future, a deep learning algorithm could be applied to quantitatively evaluate the degree of obscuration and analyze the relationship between device type and diagnostic ability in detail. Despite these limitations, we have proven for the first time that TC-cSO imaging is highly effective for diagnosing retinal diseases in patients with AH.

In conclusion, TC-cSO can diagnose retinal diseases with high accuracy and sensitivity in patients with AH. With its high spatial resolution and superior confocal effect on asteroid bodies, TC-cSO can serve as an alternative screening tool when asteroid bodies complicate the diagnosis of concurrent retinal diseases.

## Supporting information

**S1 Fig. Flow diagram for patients with asteroid hyalosis who underwent fundus photographs with three imaging devices.** AH, Asteroid hyalosis; BC-cSLO, Bicolor confocal scanning laser ophthalmoscope; TC-cSO, True-color confocal scanning ophthalmoscope; UWF, ultra-widefield. * = According to the Lens Opacities Classification System III classification. (TIF)

**S1 Table. Differences in the diagnostic ability of each imaging device according to the grade of obscuration in color fundus camera.** BC-cSLO, Bicolor confocal scanning laser ophthalmoscope; CFC, Color fundus camera; TC-cSO, True-color confocal scanning ophthalmoscope. * Grade of obscuration was defined as minimal for Grade 1 and significant for Grades 2–4. (DOCX)

## Author Contributions

**Conceptualization:** Ji Hyoung Chey, Sung Who Park, Iksoo Byon, Han Jo Kwon.

**Data curation:** Su Hwan Park, Ji Hyoung Chey, Jun Heo, Kwang Eon Han, Sung Who Park, Han Jo Kwon.

**Formal analysis:** Ji Hyoung Chey, Jun Heo, Kwang Eon Han, Han Jo Kwon.

**Investigation:** Su Hwan Park, Han Jo Kwon.

**Methodology:** Ji Hyoung Chey, Sung Who Park, Iksoo Byon, Han Jo Kwon.

**Software:** Han Jo Kwon.

**Validation:** Su Hwan Park, Han Jo Kwon.

**Visualization:** Han Jo Kwon.

**Writing – original draft:** Su Hwan Park, Ji Hyoung Chey, Sung Who Park, Iksoo Byon, Han Jo Kwon.

**Writing – review & editing:** Su Hwan Park, Ji Hyoung Chey, Iksoo Byon, Han Jo Kwon.

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
