## [Decision Letter · Decision Letter 0]

25 Apr 2024

PONE-D-23-38356Diagnostic Ability of Confocal Scanning Ophthalmoscope for the Detection of Concurrent Retinal Disease in Eyes with Asteroid HyalosisPLOS ONE

Dear Dr. Kwon,

Thank you for submitting your manuscript to PLOS ONE. After careful consideration, we feel that it has merit but does not fully meet PLOS ONE’s publication criteria as it currently stands. Therefore, we invite you to submit a revised version of the manuscript that addresses the points raised during the review process.

This research adds valuable information on the diagnosis of retina diseases in the presence of asteroid hylaosis and shows the superiority of TC-cSO imaging to the other imaging devices. The research is retrospective, but statistically analysis has been well conducted. I will suggest that the authors should include in the manuscript, why the specific statistical analysis was used and the merits of using them. Secondly, there is the issue of sample size which was raised by the reviewers. This should be addressed by the authors.  Other minor issues pointed out below should also be addressed including making Table 4 clearer.

We look forward to receiving your revised manuscript.

Kind regards,

Ogugua Ndubuisi Okonkwo, M.D.

Academic Editor

PLOS ONE

Additional Editor Comments (if provided):

Table 4 is not clear and requires more detail into how the numbers were derived. Please clarify this table for ease of understanding.

Reviewers' comments:

Reviewer's Responses to Questions

**Comments to the Author**

1. Is the manuscript technically sound, and do the data support the conclusions?

Reviewer #1: Yes

Reviewer #2: Yes

2. Has the statistical analysis been performed appropriately and rigorously? 

Reviewer #1: Yes

Reviewer #2: Yes

3. Have the authors made all data underlying the findings in their manuscript fully available?

Reviewer #1: Yes

Reviewer #2: Yes

4. Is the manuscript presented in an intelligible fashion and written in standard English?

Reviewer #1: Yes

Reviewer #2: Yes

5. Review Comments to the Author

Reviewer #1: The research is technically sound and the manuscript was written, results presented in an understandable manner.

This research sought to find out the effectiveness and precision of the confocal scanning ophthalmoscope in identifying coexisting retinal diseases in eyes with asteroid hyalosis (AH). It compared the ultrawide-field (UWF) bicolor confocal scanning laser ophthalmoscope (BC-cSLO); true color confocal scanning ophthalmoscope (TC-cSO) and color fundus camera (CFC) to a reference diagnosis gotten during a dilated fundoscopy exam. TC-cSO was found to be able to diagnose coexisting retinal diseases with high accuracy and sensitivity in patients with AH.

The strength of this study was the good interreader reliability (IRR).The weakness of the study is the small sample size and the limited range of scope of retinal diseases studied.

In the 1st paragraph of the result section, it was stated that 100 eyes met the inclusion and exclusion criteria, of which 76 (76% ) of these eyes had concurrent retinal disease coexisting with asteroid hyalosis. However in table 1 the numbers beneath the sub heading "Concurrent Retinal Disease (Ne, %)" are not adding up, they seem to have exceeded 76 eyes, please clarify.

Reviewer #2: It is a good study of retinal imaging through 3 devices in eyes with asteroid hyalosis (AH). UWF images with Optos and Eidon in eyes with AH are good

1. Can you explain any specific reason for the sample size of 100 eyes chosen?

2. How many individuals had both eyes asteroid hyalosis and how many had 1 eye alone with AH? if both eyes had AH- was 1 of the eyes chosen for the study?

3.

6. PLOS authors have the option to publish the peer review history of their article (what does this mean?). If published, this will include your full peer review and any attached files.

Reviewer #1: **Yes: **Dr. Chineze Thelma Agweye

Reviewer #2: No

---

## [Author Response · Author response to Decision Letter 0]

28 May 2024

Dear Editor & Reviewers,

We are very grateful to answer the comments about this manuscript provided by the editor and each of the reviewers. We appreciate all suggestions and corrections. All authors have carefully reviewed these comments. Our detailed responses to comments are addressed below.

Academic Editor Comments

This research adds valuable information on the diagnosis of retina diseases in the presence of asteroid hylaosis and shows the superiority of TC-cSO imaging to the other imaging devices. The research is retrospective, but statistically analysis has been well conducted. I will suggest that the authors should include in the manuscript, why the specific statistical analysis was used and the merits of using them. Secondly, there is the issue of sample size which was raised by the reviewers. This should be addressed by the authors. Other minor issues pointed out below should also be addressed including making Table 4 clearer.

Response:

The confusion matrix was employed for conducting specific analysis to evaluate diagnostic ability. The utilization of the confusion matrix for performance assessment offers several advantages. It provides a detailed breakdown of classification outcomes, including true positives, true negatives, false positives, and false negatives, enabling a deeper understanding of performance. Additionally, metrics derived from the confusion matrix such as accuracy, precision, sensitivity, and F1 score offer a comprehensive evaluation of the effectiveness across various classes or categories.

In the Method section on page 8, we included the benefits of the analysis approach for comparing performance across individual devices, stated as follows:

“Using the confusion matrix for performance evaluation offers detailed classification outcomes, while deriving key metrics like accuracy, precision, sensitivity, and F1 score provides a comprehensive evaluation of performance across different classes. Diagnostic and screening abilities for concurrent retinal diseases ~”

The questions regarding the sample size of 100 eyes and the concurrent retinal disease of 76 eyes raised by Reviewer #1 and Reviewer #2, as well as the editor's comments on clarifying Table 4, have been addressed and revised below according to each comment section.

Additional Editor Comments (if provided):

Table 4 is not clear and requires more detail into how the numbers were derived. Please clarify this table for ease of understanding.

Response:

Thank you for valuable review. As you pointed out, to describe the numbers in the table, the manuscript and the legend of the table have been modified. Table 4 presents a comparison of diagnostic ability evaluation metrics derived from the confusion matrix to determine classification accuracy for three devices.

Four ophthalmologists diagnosed six retinal diseases using fundus photographs of 100 eyes for three devices. The diagnostic performance was assessed by creating a confusion matrix for each device. The confusion matrix is composed of True Positive (TP), True Negative (TN), False Positive (FP), and False Negative (FN) values.

In confusion matrix of Each disease, total number = TP + TN + FP + FN = 4 ophthalmologists × 100 eyes = 400

In confusion matrix of All diseases, total number = TP + TN + FP + FN = 4 ophthalmologists × 100 eyes × 6 diseases= 2,400

In the results section, the explanation of Table 4 has been modified as follows:

"The four ophthalmologists diagnosed six concurrent retinal diseases using 100 fundus photographs for three imaging devices (CFC, BC-cSLO, and TC-cSLO), and the diagnostic performance of each imaging device was evaluated using confusion matrices. Table 4 summarizes the diagnostic capabilities of each imaging device."

The legend of table 4 have been added to explain the numbers in the table as follows:

“TP, TN, FP, and FN represent the values of the confusion matrix for each imaging device and disease. TP refers to the number of positive cases that were correctly identified. TN refers to the number of negative cases that were correctly identified. FP refers to the number of negative cases that were incorrectly identified as positive. FN refers to the number of positive cases that were incorrectly identified as negative.

Accuracy (%) = (TP + TN) / (TP + TN + FP + FN) × 100

Sensitivity (%) = TP / (TP + FN) × 100

Specificity (%) = TN / (TN + FP) × 100

Precision (%) = TP / (TP + FP) × 100

F1 score = 2 × (Precision × Sensitivity) / (Precision + Sensitivity)”

Additionally, I have moved the "All Diseases" column to the last position in Table 4 to prevent confusion with the six retinal diseases.

Also, the Method section on page 8 provides a detailed explanation of how accuracy and sensitivity were calculated using the results from the confusion matrix.

“Diagnostic and screening abilities for concurrent retinal diseases in patients with AH were investigated using the following four-step process: First, based on the medical records and three fundus images, the corresponding author diagnosed concurrent retinal disease for each retinal image as a reference diagnosis. The corresponding author created a list of potential retinal diseases including DR, dry and wet age-related macular degeneration (AMD), retinal vein occlusion (RVO), epiretinal membrane (ERM), and retinitis pigmentosa. Third, four independent ophthalmologists (S.H.P., J.H.C., J.H., and K.E.H.) recorded possible retinal diseases accompanying AH as diagnostic tests, based solely on fundus photographs taken with CFCs, UWF BC-cSLOs, and TC-cSOs. The diagnostic test was continued until each ophthalmologist interpreted all fundus photographs. Finally, the corresponding author compared the diagnoses of the four ophthalmologists against the reference diagnosis and constructed confusion matrices for each ophthalmologist's assessment. These matrices were aggregated to calculate the accuracy and sensitivity of each imaging device for each disease and to evaluate their diagnostic capabilities.”

Reviewer #1:

The research is technically sound and the manuscript was written, results presented in an understandable manner.

This research sought to find out the effectiveness and precision of the confocal scanning ophthalmoscope in identifying coexisting retinal diseases in eyes with asteroid hyalosis (AH). It compared the ultrawide-field (UWF) bicolor confocal scanning laser ophthalmoscope (BC-cSLO); true color confocal scanning ophthalmoscope (TC-cSO) and color fundus camera (CFC) to a reference diagnosis gotten during a dilated fundoscopy exam. TC-cSO was found to be able to diagnose coexisting retinal diseases with high accuracy and sensitivity in patients with AH.

The strength of this study was the good interreader reliability (IRR).The weakness of the study is the small sample size and the limited range of scope of retinal diseases studied.

In the 1st paragraph of the result section, it was stated that 100 eyes met the inclusion and exclusion criteria, of which 76 (76% ) of these eyes had concurrent retinal disease coexisting with asteroid hyalosis. However in table 1 the numbers beneath the sub heading "Concurrent Retinal Disease (Ne, %)" are not adding up, they seem to have exceeded 76 eyes, please clarify.

Response:

Thank you for the correction.

In Table 1, a correction has been made from RP 10 eyes to 3 eyes due to a typographical error, with no impact on other results in the paper. Consequently, Concurrent retinal disease is as follows: DR 20 eyes, Dry AMD 31 eyes, Wet AMD 9 eyes, RVO 6 eyes, ERM 33 eyes, and RP 3 eyes.

Furthermore, when there are two or more retinal diseases present in eyes with DR, Dry AMD, Wet AMD, RVO, ERM, and RP, there are 22 eyes; when there are two retinal diseases, there are 18 eyes; and when there are three retinal diseases, there are 4 eyes. To avoid potential confusion arising from the discrepancy in the total count when labeling cases with two or more retinal diseases as "Multiple retinal diseases" in Table 1, we opted to differentiate between cases with two retinal diseases and those with three retinal diseases.

Upon recalculating, if we add up all concurrent retinal diseases, including overlaps, the total number of individual cases would be 20 + 31 + 9 + 6 + 33 + 3 = 102 cases. To calculate the total number of eyes having concurrent retinal disease, we must subtract those overlaps (cases with multiple retinal diseases): 102 - (18 × 1) - (4 × 2) = 76 eyes. Therefore, the number is correct.

Additionally, to aid comprehension, the following explanation has been added to the legend and footnote in Table 1: “Two retinal diseases and Three retinal diseases represent the number of eyes diagnosed with multiple diseases among the listed concurrent retinal disease (DR, Dry AMD, Wet AMD, RVO, ERM, and RP).”

Reviewer #2:

It is a good study of retinal imaging through 3 devices in eyes with asteroid hyalosis (AH). UWF images with Optos and Eidon in eyes with AH are good

1. Can you explain any specific reason for the sample size of 100 eyes chosen?

Response:

After excluding eyes with unsatisfactory fundus photography quality from a pool of 101 patients (119 eyes), the dataset was narrowed down to 89 patients (100 eyes). It's important to note that the selection was not deliberately targeted at 100 eyes; rather, the inclusion criterion led to 100 eyes being included in the analysis.

The Method section on page 6 outlines the exclusion criteria as described below, along with an explanation in the flow diagram depicted in S1 Figure.

“Patients with conditions other than AH that could affect the quality of fundus photography, such as corneal opacity, grade ≥4 nuclear opacity, or grade ≥2 posterior subcapsular opacity according to the Lens Opacities Classification System III, or vitreous hemorrhage were not included in the study. Additionally, patients with retinal or choroidal detachment that interfered with focusing on the entire macula, those whose pupils did not dilate to at least 4 mm under low light (5 lx), and those with involuntary eye movements causing overlapping images were excluded from the study.”

2. How many individuals had both eyes asteroid hyalosis and how many had 1 eye alone with AH? if both eyes had AH- was 1 of the eyes chosen for the study?

Response:

89 patients (100 eyes) were included. 11 patients (12.4%) had bilateral involvement, while 78 patients (87.6%) had unilateral involvement. In Table 1, for the 89 patients included, the distribution is represented as Bilateral/Unilateral (Np, %) 11 (12.4%)/78 (87.6%).

When AH is present in both eyes, fundus images from both eyes were included, rather than only fundus image from one eye.

---

## [Editor Report · Decision Letter 1]

11 Jun 2024

Diagnostic Ability of Confocal Scanning Ophthalmoscope for the Detection of Concurrent Retinal Disease in Eyes with Asteroid Hyalosis

PONE-D-23-38356R1

Dear Dr. Kwon,

We’re pleased to inform you that your manuscript has been judged scientifically suitable for publication and will be formally accepted for publication once it meets all outstanding technical requirements.

Kind regards,

Ogugua Ndubuisi Okonkwo, M.D.

Academic Editor

PLOS ONE
---

## [Editor Report · Acceptance letter]

25 Nov 2024

PONE-D-23-38356R1 

PLOS ONE

Dear Dr. Kwon, 

I'm pleased to inform you that your manuscript has been deemed suitable for publication in PLOS ONE. Congratulations! Your manuscript is now being handed over to our production team.

Kind regards, 

on behalf of

Dr. Ogugua Ndubuisi Okonkwo 

Academic Editor

PLOS ONE